# Effects of Sex on the Muscle Development and Meat Composition in Wuliangshan Black-Bone Chickens

**DOI:** 10.3390/ani12192565

**Published:** 2022-09-26

**Authors:** Zhengmiao Ou, Yanyan Shi, Qingqing Li, Yun Wu, Fenfen Chen

**Affiliations:** 1College of Life Science, Southwest Forestry University, Kunming 650224, China; 2School of Biology and Agriculture Technology, Zunyi Normal University, Zunyi 563006, China

**Keywords:** Wuliangshan Black-bone chicken, sex, breast muscle, thigh muscle, muscle hypertrophy, fatty acid, amino acid

## Abstract

**Simple Summary:**

Wuliangshan Black-bone chickens, an indigenous chicken breed in southwestern China, have characteristics of black skin, black bones, and black meat, and are suitable for rearing at high altitudes. After six generations of selection, the bodyweights of chickens have been greatly improved: cocks and hens have reached 2.0 kg and 1.5 kg weight at 126 days of age, respectively. The breast and thigh muscles are the main meat-producing parts of chicken; assessing and documenting the morphological features of breast and thigh muscle development and meat composition in Wuliangshan Black-bone chickens is very important in terms of their genetic breeding and also consumer interest.

**Abstract:**

This study was conducted to illustrate the morphological features of the breast and thigh muscles at four developmental stages (1, 42, 84, and 126 days of age) of Wuliangshan Black-bone chickens and to compare the chemical composition, fatty acid, and amino acid contents of their meat at 126 days of age (D126). In total, 80 chickens (male, *n* = 40 and female, *n* = 40) in the sixth generation from the breeding farm were used in the experiment under free-range rearing conditions. The cross-sectional areas (CSA) of muscle fibers and meat composition were compared between different sex and different muscle types. The results showed that gender did not affect the CSA of the breast muscle fibers but did affect the CSA of the thigh muscle fibers at D42, D84, and D126 (*p* < 0.05). Muscle types affected the CSA of muscle fibers: male chickens exhibited significantly higher values than female chickens at D42, D84, and D126 (*p* < 0.05). The results of moisture, crude protein, and crude fat at D126 showed that moisture contents were higher in the thigh muscles than in the breast muscles in male and female chickens (*p* < 0.05). Gender affected the crude protein contents and crude fat contents: the breast muscle crude protein content was significantly higher than that in the thigh muscle, both in males and females (*p* < 0.05), whereas the crude fat contents were significantly higher in females than in males (*p* < 0.05); moreover, the thigh muscle fat contents were significantly higher than those of the breast muscles both in males and females (*p* < 0.05). Gender and muscle types also affected the fatty acid contents: the PUFA contents of the breast and thigh muscles were significantly higher in male than in female chickens (*p* < 0.05). Muscle types significantly influenced the total EAA, NEAA, and flavor amino acid contents. The total EAA contents of the breast muscles were significantly higher than those of the thigh muscles in males and females (*p* < 0.05), whereas the total NEAA and total flavor amino acid contents of the thigh muscles were significantly higher than those of the breast muscles (*p* < 0.05). Our results may lead to a better understanding of the effects of gender on the breast and thigh muscle development and meat composition of Wuliangshan Black-bone chicken.

## 1. Introduction

The continuing demand for increased local poultry meat production has attracted the interests on muscle tissue development and meat composition related study to improve meat performance [1,2,3,4,5]. The ultimate goal is to provide high levels of growth performance and quality of their meat. The development and composition of breast and thigh muscles, as the main meat-producing parts of chickens, are affected by breed [6], sex [7], nutrition [8], intestinal microbial species [9], and rearing systems [10]. Chicken is a favorite food product for healthy eating because it contains higher levels of protein as well as lower fat content than other meats. Moreover, indigenous chicken breeds have unique flavors, compared with broilers, which are closely related to the fat contents, fatty acids, and amino acids in meat. Gender is the main influencing factor for meat quality under the same genetic backgrounds and breeding methods.

Wuliangshan Black-bone chickens, one of Yunnan’s ‘top six famous chicken’ breeds in China, are known for their high nutritional value, tender meat, and good taste; however, little is known about the different muscle-type effects on the morphological characteristics of muscle development and meat composition regarding the differences between sexes [4]. Therefore, this study examined the morphological features of breast and thigh muscle development, muscle chemical composition, and fatty acid and amino acid contents in male and female chickens. The results of this study could help conserve and characterize Wuliangshan Black-bone chickens with respect to their muscle development and meat quality traits.

## 2. Materials and Methods

### 2.1. Animals and Sample Collection

Wuliangshan Black-bone chickens were acquired from the Breeding Farm of Wuliangshan Black-bone chickens (Nanjian, China) and handled in accordance with the guidelines of the Southwest Forestry University Animal Care Committee (ethical approval number SWFU-2021019). The chickens were allowed access to feed and water ad libitum under the same feedstuff and management conditions and were humanely sacrificed at the end of the experiment to ameliorate suffering. All the chickens were sacrificed via CO_2_ asphyxiation. The breast muscle and thigh muscle samples were collected from twenty chickens (ten male and ten female) on day 1 (D1), D42, D84, and D126. To determine the meat composition and HE staining, the breast muscle (*pectoralis major*) and the thigh muscle (*peroneus longus*) from the left side were collected, HE staining samples were fixed with 4% paraformaldehyde, and other samples were immediately frozen in liquid nitrogen and then transported to the laboratory and stored at −80℃ for further analysis.

### 2.2. Hematoxylin–Eosin (HE) Staining and Measurement of the Muscle Cell Size

The breast and thigh muscle samples were fixed with 4% paraformaldehyde and kept at room temperature. The fixed tissues were dehydrated in 30% sucrose (*v/v*) and sectioned (5 μm) through a sliding microtome (Leica, Solms, Germany). The sections were stained with HE using standard pathologic procedures. ImageJ software (NIH) was used to measure the muscle cell size. The average cross-sectional area (CSA) of myofiber was performed according to a previously published method [11]. Over 100 bundles of myofiber were measured for each animal.

### 2.3. Meat Chemical Composition

The moisture, crude protein, and crude fat compositions of breast and thigh meat from the chickens were determined following the AOAC (2009) standard method. The breast and thigh muscles from chickens were isolated, and all the visible connective tissue was removed. Moisture was detected with an oven-drying method; crude proteins were identified using the Kjeldahl method, while crude fats were assessed using the Soxhlet extractor method.

### 2.4. Fatty Acid Content Determination

The fatty acid contents were determined according to a previously published method [12]. The breast and thigh muscle samples were treated with concentrated hydrochloric acid and then a petroleum ether/diethyl ether mixture for total lipid extraction. The fatty acids were analyzed as their methyl esters, and transesterification was performed using a 15% solution of boron trifluoride in methanol. The prepared samples were then determined using gas chromatography equipped with a flame-ionization detector (Gas Chromatograph GC, Thermo trace 1300, Shanghai, China) with a capillary column Agilent J&W (CP-sil88, NL) and helium as the carrier gas. The quantification was carried out through area normalization with an external standard mixture of fatty acid methyl esters (TMRM, 81817, China). The fatty acid composition was calculated as a percentage of each individual fatty acid relative to the total fatty acids.

### 2.5. Amino Acid Content Determination

The amino acid contents were analyzed using a Biochrom30+ Automatic Amino Acid Analyzer (Dachang Huajia (Shanghai) International Trade Co., Ltd., Beijing, China). The values were determined from the breast muscle and thigh muscle using a 0.2 g meat sample and 10 mL 6mol/L hydrochloric acid as the ampoule. The ampoule was frozen in a refrigerator at −20 °C for 3–5min, filled with nitrogen, sealed with an alcohol spray lamp, and placed in a thermostatic drying chamber at 110 ± 1 °C for hydrolysis for 22 h. After cooling to room temperature, the hydrolysate was transferred to a 50mL volumetric flask, rinsed with a small amount of deionized water several times, and the water washing solution was transferred to the volumetric flask together, diluted to a constant volume, and shaken evenly. Subsequently, 2.0 mL of the filtrate was accurately absorbed and placed in an evaporating dish maintained at 60 °C in a water bath for drying. When the liquid was rapidly evaporated, by adding 2.0 mL of deionized water, the procedure was repeated three times until evaporation. Then, 2.0 mL of a pH 2.2 diluent was added, mixed well, passed through a 0.45 μm filter membrane, and transferred to the instrument injection bottle for sample determination. FAs were quantified using the internal standard after adjusting for the response, as determined by using AAS18-5ML Amino Acid Standard mixtures (Dachang Huajia (Shanghai) International Trade Co., Ltd., Beijing, China).

### 2.6. Statistical Analysis

GraphPad Prism 8.0 was utilized for visualization. The data were analyzed with SPSS 22.0 software (SPSS Science, Chicago, IL, USA), and the differences between the groups were evaluated using unpaired Student’s *t*-tests. The data are presented as the mean ± SEM. A value of *p* < 0.05 was considered to be statistically significant.

## 3. Results

### 3.1. Histological Comparisons of Breast and Thigh Muscles in Wuliangshan Black-Bone Chickens

The breast and thigh muscles were observed via HE staining; *t*-tests were performed to compare the differences in the cross-sectional areas (CSAs) of the muscle fibers between male and female chickens at D1, D42, D48, and D126 (Figure 1). The current study revealed that, in the same gender, the thigh muscle exhibited a larger myofiber area, compared with the breast muscle at D1 and D126 (*p* < 0.05) (Figure 1B,E); moreover, the same result was demonstrated in males at D84 (*p* < 0.05) (Figure 1C). In the thigh muscle, the CSA of the muscle fibers of male chickens was significantly higher than that of female chickens at D42, D84, and D126 (*p* < 0.05) (Figure 1C–E). In breast muscle, there were no differences in the CSA of the muscle fibers between male and female chickens (Figure 1B–E). Therefore, these results indicated that muscle type did affect the CSA of muscle fibers.

### 3.2. Chemical Composition of Breast and Thigh Muscles at D126 in Wuliangshan Black-Bone Chickens

The moisture, crude protein, and crude fat at D126 of the breast and thigh muscles are detailed in Table 1. Moisture contents of thigh muscle were higher than in the breast muscle in both male and female chickens. Males breast muscle showed higher crude protein contents than females (*p* < 0.05). Whereas breast muscle crude protein content was significantly higher than that in thigh muscle both in males and females (*p* < 0.05). Muscle type and gender affected crude fat content: the thigh muscle fat contents were significantly higher than those in breast muscles in both males and females, whereas the crude fat contents of breast and thigh muscles were significantly higher in females than in males (*p* < 0.05).

### 3.3. Fatty Acid Composition of Breast and Thigh Muscles at D126 in Wuliangshan Black-Bone Chickens

The fatty acid compositions of breast and thigh meat at D126 of Wuliangshan Black-bone chickens are summarized in Table 2. In the breast muscle, the levels of C13:0, C15:0, C16:0, C20:0, C18:1(n-9)t, and C20:3(n-6) in female chickens were significantly higher than those in male chickens (*p* < 0.05), whereas the levels of C18:3(n-6), C20:2(n-6), C20:4(n-6), C22:3(n-3), Ʃn-6, Ʃn-3, and ƩMUFA in male chickens were significantly higher than those in female chickens (*p* < 0.05). In the thigh muscle, the levels of C14:0, C22:0, C16:1, C18:1(n-9)t, C18:1(n-9)c, C20:3(n-6), C18:3(n-3), C18:3(n-6), ƩMUFA, and ƩMUFA/ƩSFA in female chickens were significantly higher than those in male chickens (*p* < 0.05), whereas the levels of C18:0, C20:4(n-6), C22:3(n-3), ƩSFA, Ʃn-6, and ƩPUFA in male chickens were significantly higher than those in female chickens (*p* < 0.05).

The results of comparing the fatty acid compositions of breast muscle and thigh muscles in the same genders are also shown in Table 2. In male chickens, the levels of C15:0, C16:0, C24:0, C18:3(n-6), C20:2(n-6), C20:3(n-6), C20:4(n-6), C20:3(n-3), C22:3(n-3), Ʃn-3, and ƩPUFA in the breast muscles were significantly higher than those in the thigh muscles (*p* < 0.05), whereas the levels of C14:0, C17:0, C20:0, C22:0, C14:1, C20:1, C18:2(n-6), C18:3(n-3), and Ʃn-6/Ʃn-3 in the thigh muscles were significantly higher than those in the breast muscles (*p* < 0.05). In female chickens, the levels of C16:0, C24:0, C20:2(n-6), C20:3(n-6), C20:4(n-6), C20:3(n-3), C20:3(n-3), ƩSFA, Ʃn-3, and ƩPUFA in the breast muscles were significantly higher than in the thigh muscles (*p* < 0.05), whereas the levels of C14:0, C15:0, C14:1, C16:1, C18:1(n-9)c, C20:1, C18:2(n-6), C18:3(n-3), C18:3(n-3), ƩMUFA, ƩMUFA/ƩSFA, and Ʃn-6/Ʃn-3 in the thigh muscles were significantly higher than in the breast muscles (*p* < 0.05). The results showed that the compositions of MUFA were lower and the compositions of PUFA were higher in Wuliangshan Black-bone chickens.

### 3.4. Amino Acid Composition of Breast and Thigh Muscles at D126 in Wuliangshan Black-Bone Chickens

The results of the amino acid composition of the breast and thigh muscles at D126 are depicted in Table 3. In the breast muscle, the levels of Leu, Lys, Ala, and Tyr in male chickens were significantly higher than the levels in female chickens (*p* < 0.05), whereas the levels of Met, Phe, Cys, and His in female chickens were significantly higher than those in male chickens (*p* < 0.05). In the thigh muscle, the levels of Phe, Thr, and Tyr in male chickens were significantly higher than those in female chickens (*p* < 0.05).

Muscle type significantly influenced the total EAA, NEAA, and flavor amino acid contents (Table 3). The total EAA contents of the breast muscles were significantly higher than the contents in the thigh muscles in males and females, whereas the total NEAA and total flavor amino acid contents in the thigh muscles were significantly higher than the contents in the breast muscles.

## 4. Discussion

Breeding for local chicken breeds has led to increased growth performance and maintained meat quality. Sex differences are associated with growth performance in chickens [9], and genetic studies have reported that muscle characteristics, such as fiber CSA, have high heritability (>0.40) and exhibit significant positive genetic correlations with body and muscle weights in broilers between 4 and 6 weeks [13]. Moreover, muscle morphology is affected by many factors, such as breed [14], muscle type [10], age [11,13,15], and gene polymorphism [11]. In the present study, the muscle morphological characteristics of the breast and thigh muscles were assessed at different developmental stages, and the CSA of the breast and thigh muscles increased with age. There were no differences between males and females in the CSA of the breast muscle, but the CSAs of the thigh muscle fibers in cocks were significantly higher than those in hens at D42, D84, and D126. This is possibly because the breast muscle develops rapidly after birth, which causes the gradual narrowing of the gap between the breast and the thigh muscle fiber. However, with increases in age and levels of exercise, the gap between them gradually increased. Previous reports have suggested that gender is not the main factor affecting the CSA in the same muscle types [16]. However, in this study, the CSAs of the thigh muscles in cocks were higher than those in hens, indicating that gender affected the CSA of the thigh muscles. This was possibly due to the chickens being reared in a floor rearing system, and the cocks had higher levels of exercise than the hens, which made their muscle fibers coarser.

The chemical composition of chicken meat is closely related to breeding, which means that genetics is the main influencing factor [1,17,18]. In this study, we found large moisture difference between breast muscle and thigh muscle, but not genders. Compared with female, male exhibited higher breast muscle crude protein contents than thigh meat, but not the crufe fat content. A recent study reported that two Italian slow-growing chicken breeds also exhibited the same results [7]. Another study reported that the higher crude fat contents in female meat contributed to estrogen production, promoting lipid synthesis and deposition during the sexual maturity of birds [19].

Poultry meat is rich in essential PUFAs, especially ω-3 fatty acids [14,15]. As an excellent local breed, Wuliangshan Black-bone chickens characteristic of a high nutritional value, tender meat, and good taste [4]. Our results showed that the breast muscle had higher levels of polyunsaturated fatty acids, especially n-3 polyunsaturated fatty acids. Also, the contents of the essential fatty acids in breast meat were higher than those in the thigh meat. These results are in agreement with those of Mahiza et al., which reported that the compositions of MUFA were lower and the compositions of PUFA were higher in slow-growing birds than in fast-growing birds [20].

The amino acid composition in the muscle is an important factor affecting poultry quality, and the characteristic ‘poultry’ flavor is closely related to it [15]. The current study revealed that there were no significant differences in the amino acid compositions between the sexes. The total amounts of essential amino acids were higher in the breast muscle than in the thigh muscles, but the levels of flavor amino acids were higher in the thigh muscle than in the breast muscles.

## 5. Conclusions

It is concluded that gender has a significant effect on thigh muscle, but not on breast muscle; In terms of muscle fiber area, gender has a significant impact on thigh muscle.Both gender and muscle types affected the chemical contents and fatty acid and amino acid compositions of meat at D126 in Wuliangshan Black-bone chickens. From a nutritional point of view, the characteristics of the meat from the thigh muscles are better than breast muscles.

## Figures and Tables

**Figure 1 animals-12-02565-f001:**
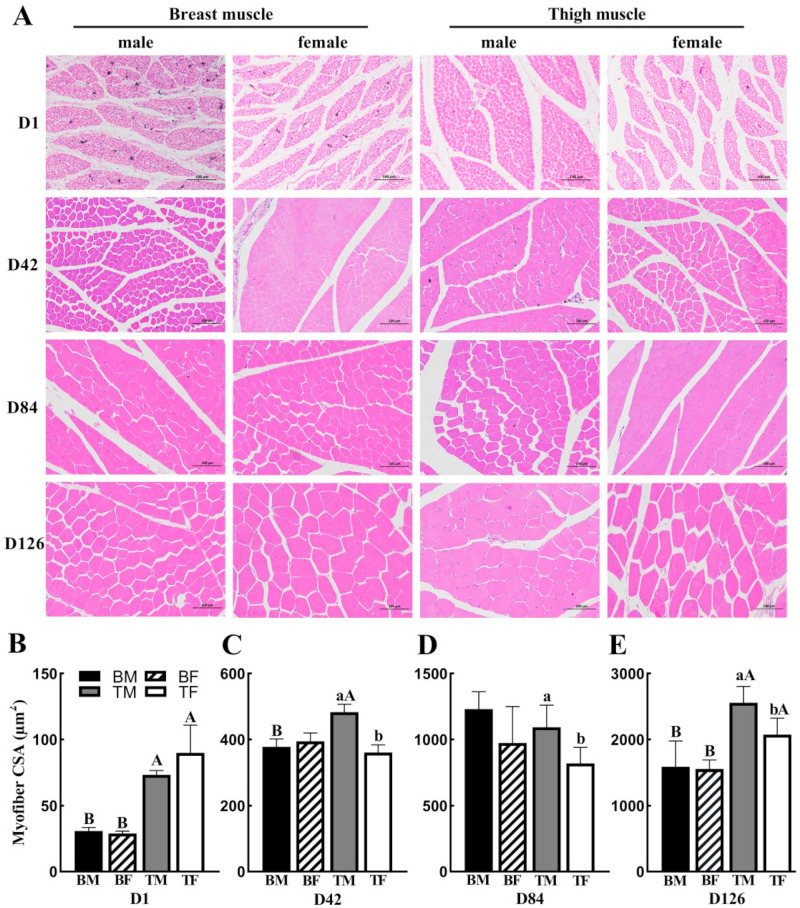
Morphological changes in breast and thigh muscle at D1, D42, D84, and D126 between male and female Wuliangshan Black-bone chickens: (**A**) HE (hematoxylin–eosin) staining of breast and thigh muscles. Scale bar: 100 μm for D1, D42, D84, and D126; (**B**–**E**) comparisons of the CSA of breast and thigh muscle fibers. Different lowercase letters on each bar indicate the same type of muscle; there are significant differences (*p* < 0.05) between genders. Different capital letters on each bar indicate the same sex; there are significant differences (*p* < 0.05) between breast and thigh muscles. Abbreviations: CSA, cross-sectional area; D, postnatal days; BM, breast muscle of male; BF, breast muscle of female; TM, thigh muscle of male; TM, thigh muscle of male.

**Table 1 animals-12-02565-t001:** Chemical composition of breast and thigh muscles of Wuliangshan Black-bone chickens (g/100g FM).

Item	Breast Muscle		Thigh Muscle	
	Male	Female	Male	Female
Moisture	74.42 ± 0.17	74.38 ± 0.18	76.31 ± 0.26 ^A^	76.31 ± 0.16 ^A^
Crude protein	23.64 ± 0.75 ^aA^	22.89 ± 0.17 ^bA^	20.29 ± 0.35 ^B^	19.96 ± 0.19 ^B^
Crude fat	0.68 ± 0.02 ^bB^	0.92 ± 0.18 ^aB^	1.15 ± 0.09 ^bA^	1.78 ± 0.16 ^aA^

Abbreviations: FM, fresh matter; N.D, not detected. Values with different lowercase letters in the same line indicate the same type of muscle; there are significant differences (*p* < 0.05) between genders. Values with different capital letters in the same line indicate the same sex; there are significant differences (*p* < 0.05) between breast and thigh muscles (*n* = 10).

**Table 2 animals-12-02565-t002:** The contents of fatty acids in breast and thigh muscles of Wuliangshan Black-bone chickens (%).

Items #	Breast Muscle		Thigh Muscle	
	Male	Female	Male	Female
C12:0	N.D.	0.01 ± 0.01	N.D.	0.01 ± 0.00
C13:0	0.02 ± 0.01 ^b^	0.03 ± 0.01 ^a^	N.D.	N.D.
C14:0	0.19 ± 0.03 ^B^	0.25 ± 0.02 ^B^	0.30 ± 0.04 ^bA^	0.47 ± 0.02 ^aA^
C15:0	0.03 ± 0.01 ^bA^	0.09 ± 0.01 ^a^	0.09 ± 0.01	0.12 ± 0.01 ^A^
C16:0	21.85 ± 0.43^bA^	23.78 ± 0.30 ^aA^	18.65 ± 0.70 ^B^	20.42 ± 2.28 ^B^
C17:0	0.19 ± 0.02 ^B^	0.19 ± 0.01	0.26 ± 0.01 ^A^	0.24 ± 0.02
C18:0	16.87 ± 0.49	14.77 ± 0.61	18.93 ± 1.41 ^a^	12.75 ± 0.86 ^b^
C20:0	0.05 ± 0.001 ^bB^	0.08 ± 0.01 ^a^	0.12 ± 0.01 ^A^	0.11 ± 0.00
C22:0	0.03 ± 0.01 ^B^	0.04 ± 0.00	0.06 ± 0.00 ^bA^	0.04 ± 0.01 ^a^
C24:0	0.16 ± 0.02 ^A^	0.17 ± 0.01 ^A^	0.08 ± 0.01 ^B^	0.07 ± 0.01 ^B^
**∑SFA**	39.48 ± 0.27	39.60 ± 0.54 ^A^	38.52 ± 0.90 ^a^	34.26 ± 1.99 ^bB^
C14:1	0.02 ± 0.02 ^B^	0.04 ± 0.02 ^B^	0.05 ± 0.01 ^A^	0.07 ± 0.01 ^A^
C16:1	0.20 ± 0.05	0.60 ± 0.13 ^B^	0.56 ± 0.13 ^b^	1.74 ± 0.90 ^aA^
C18:1(n-9)t	0.01 ± 0.01 ^b^	0.06 ± 0.01 ^a^	0.03 ± 0.01 ^b^	0.07 ± 0.01 ^a^
C18:1(n-9)c	17.66 ± 0.72	20.62 ± 1.06 ^B^	20.53 ± 1.51 ^b^	27.88 ± 1.18 ^aA^
C20:1	0.19 ± 0.02 ^B^	0.16 ± 0.01 ^B^	0.28 ± 0.02 ^A^	0.25 ± 0.03 ^A^
**∑MUFA**	18.08 ± 0.77	21.47 ± 1.18 ^B^	21.45 ± 1.63 ^b^	30.01 ± 4.53 ^aA^
C18:2(n-6)	18.77 ± 1.04 ^B^	19.78 ± 0.58 ^B^	25.69 ± 0.94 ^A^	27.10 ± 1.16 ^A^
C18:3(n-6)	0.10 ± 0.0 ^aA^	0.05 ± 0.01 ^bB^	0.05 ± 0.01 ^bB^	0.07 ± 0.01 ^aA^
C20:2(n-6)	0.90 ± 0.06 ^aA^	0.67 ± 0.08 ^bA^	0.61 ± 0.04 ^bB^	0.37 ± 0.04 ^bB^
C20:3(n-6)	0.60 ± 0.03 ^bA^	0.82 ± 0.06 ^aA^	0.33 ± 0.02 ^B^	0.33 ± 0.04 ^B^
C20:4(n-6)	18.59 ± 1.31 ^aA^	14.57 ± 1.01 ^bA^	11.34 ± 1.39 ^aB^	5.94 ± 0.86 ^bB^
**∑n-6**	38.77 ± 0.51 ^a^	35.89 ± 0.69 ^b^	38.01 ± 0.96 ^a^	33.82 ± 1.51 ^b^
C18:3(n-3)	0.32 ± 0.09 ^B^	0.53 ± 0.09 ^B^	0.65 ± 0.14 ^bA^	1.150 ± 0.10 ^aA^
C20:3(n-3)	0.06 ± 0.01 ^A^	0.05 ± 0.01 ^A^	0.03 ± 0.00 ^B^	0.02 ± 0.00 ^B^
C22:3(n-3)	3.29 ± 0.26 ^aA^	2.48 ± 0.16 ^bA^	1.35 ± 0.21 ^aB^	0.75 ± 0.12 ^bB^
**∑n-3**	3.67 ± 0.20 ^aA^	3.06 ± 0.15 ^bA^	2.02 ± 0.13 ^B^	1.91 ± 0.12 ^B^
**∑PUFA**	42.44 ± 0.63 ^aA^	38.96 ± 0.78 ^bA^	40.03 ± 0.97 ^aB^	35.73 ± 1.63 ^bB^
**∑MUFA/∑SFA**	0.46 ± 0.02	0.55 ± 0.04 ^B^	0.57 ± 0.05 ^b^	0.94 ± 0.13 ^aA^
**∑PUFA/∑SFA**	1.08 ± 0.01	0.98 ± 0.02	1.04 ± 0.03	1.13 ± 0.18
**∑n-6/∑n-3**	10.82 ± 0.55 ^B^	11.95 ± 0.58 ^B^	19.50 ± 1.32 ^A^	18.16 ± 1.08 ^A^

Abbreviations: SFA, saturated fatty acid; MUFA, monounsaturated fatty acid; PUFA, polyunsaturated fatty acid; N.D., not detected. Values with different lowercase letters in the same line indicate the same type of muscle; there are significant differences (*p* < 0.05) between genders. Values with different capital letters in the same line indicate the same sex; there are significant differences (*p* < 0.05) between breast and thigh muscles. # The items displayed in the table were identified from 36 kinds of fatty acids with detectable rates higher than 30%; those otherwise are marked with N.D.

**Table 3 animals-12-02565-t003:** The contents of amino acids in breast and thigh muscles of Wuliangshan Black-bone chickens (%).

Items	Breast Muscle		Thigh Muscle	
	Male	Female	Male	Female
Arg *	6.88 ± 0.01 ^B^	6.84 ± 0.20 ^B^	7.09 ± 0.02 ^A^	7.11 ± 0.02 ^A^
Ile	5.14 ± 0.01 ^A^	5.11 ± 0.03 ^A^	4.97 ± 0.02 ^B^	4.90 ± 0.04 ^B^
Leu	8.31 ± 0.04 ^aB^	8.24 ± 0.05 ^bA^	8.15 ± 0.02 ^A^	8.08 ± 0.04 ^B^
Lys	9.19 ± 0.01 ^aA^	9.05 ± 0.03 ^b^	9.07 ± 0.04 ^B^	9.03 ± 0.03
Met	2.78 ± 0.04 ^b^	2.97 ± 0.03 ^aA^	2.68 ± 0.05	2.81 ± 0.05 ^B^
Phe	5.45 ± 0.01 ^bA^	5.53 ± 0.02 ^aA^	4.84 ± 0.03 ^aB^	4.76 ± 0.02 ^bB^
Thr	4.79 ± 0.01	4.75 ± 0.02 ^A^	4.75 ± 0.01 ^a^	4.68 ± 0.02 ^bB^
Val	5.33 ± 0.02 ^A^	5.31 ± 0.03 ^A^	4.99 ± 0.03 ^B^	4.93 ± 0.04 ^B^
∑Eaa	47.88 ± 0.07 ^A^	47.81 ± 0.03 ^A^	46.55 ± 0.09 ^B^	46.29 ± 0.19 ^B^
Ala *	6.08 ± 0.02 ^aB^	6.02 ± 0.02 ^bB^	6.15 ± 0.02 ^A^	6.16 ± 0.04 ^A^
Asp *	9.45 ± 0.02 ^A^	9.45 ± 0.02 ^A^	9.24 ± 0.02 ^B^	9.28 ± 0.03 ^B^
Cys	0.17 ± 0.02 ^bB^	0.26 ± 0.03 ^a^	0.25 ± 0.02 ^A^	0.27 ± 0.02
Glu *	15.60 ± 0.04 ^aB^	15.44 ± 0.04 ^bB^	16.43 ± 0.03 ^A^	16.43 ± 0.05 ^A^
Gly *	4.55 ± 0.04 ^B^	4.55 ± 0.02 ^B^	5.30 ± 0.06 ^A^	5.57 ± 0.14 ^A^
His	4.16 ± 0.08 ^bA^	4.47 ± 0.04 ^aA^	3.15 ± 0.04 ^B^	3.09 ± 0.03 ^B^
Pro	3.92 ± 0.05 ^B^	3.88 ± 0.03 ^B^	4.55 ± 0.04 ^A^	4.68 ± 0.05 ^A^
Ser	4.21 ± 0.01 ^B^	4.17 ± 0.01 ^B^	4.39 ± 0.01 ^A^	4.34 ± 0.02 ^A^
Tyr	3.98 ± 0.01 ^a^	3.94 ± 0.01 ^bA^	3.99 ± 0.03 ^a^	3.88 ± 0.01 ^bB^
∑NEAA	52.12 ± 0.07 ^B^	52.19 ± 0.03 ^B^	53.45 ± 0.09 ^A^	53.71 ± 0.19 ^A^
∑flavor amino acid	42.56 ± 0.07 ^B^	42.31 ± 0.07 ^B^	44.22 ± 0.10 ^A^	44.55 ± 0.20 ^A^

* Indicates a flavor amino acid. Abbreviations: EAA, essential amino acid; NEAA, non-essential amino acid. Values with different lowercase letters in the same line indicate the same type of muscle; there are significant differences (*p* < 0.05) between genders. Values with different capital letters in the same line indicate the same sex; there are significant differences (*p* < 0.05) between breast and thigh muscles.

## Data Availability

Not applicable.

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
