# Peer review of "Effects of Sex on the Muscle Development and Meat Composition in Wuliangshan Black-Bone Chickens"

_animals, 2022, doi:10.3390/ani12192565_

Round 1

Reviewer 1 Report

In this manuscript, the authors aimed to illustrate the morphological features of breast and thigh muscles at four developmental stages of Wuliangshan Black-bone chickens, and to compare the chemical composition, fatty acid and amino acid content of meat at 126 days of age. The objective of this study was clearly described. Additionally the experimental procedures were well considered and conducted appropriately.

Major comments to the authors

(1)Introduction: First paragraph is scattered and lacks clarity.

(2)In materials and methods, the samples were collected from different days of age. However, fatty acids and amino acids were only examined at 126 days of age. The authors need to describe precisely in the materials and methods section.

Minor comments

p. 1 line 21-26 Abstract

It was very hard to read those sentences. Please rewrite.

p. 1 line 41-42 Keywords 

It was not the right words “muscle hypertrophy”. Please rewrite.

p. 3 line 95-98 Methods 2.4

Not clear what authors mean in line 95-98. Please rewrite.

p. 3 line 131

In materials and methods, Hematoxylin-eosin staining was presented as HE staining. However, here it was shown by H&E staining. Please keep the full text consistent with each other.

p. 6 line 197 

N.D should be spelled out. 

Author Response

We have made a careful and complete inspection throughout the manuscript. The revision in manuscript was marked by track changes and the followed words were the response for reviewer’s comments.

Reviewer 1

  1. Introduction: First paragraph is scattered and lacks clarity.

Reply: Thanks for your question. We have made some amendments in introduction (line 50-67). We hope these revisions can make the introduction more clear and understandable. Thank you so much.

  1. In materials and methods, the samples were collected from different days of age. However, fatty acids and amino acids were only examined at 126 days of age. The authors need to describe precisely in the materials and methods section.

Reply: Thanks for your question. Because the weight at 126 days of age had already reached the slaughter weight, only the meat composition at 126 days of age was examined. The explanation for the reason has provided at the simple summary (line 14-15).

  1. p. 1 line 21-26 Abstract. It was very hard to read those sentences. Please rewrite.

Reply: Thanks for your suggestion. We have made some amendments in abstract (line 19-41). We hope these revisions can make the abstract more clear and understandable. Thank you so much.

  1. p. 1 line 41-42 Keywords. It was not the right words “muscle hypertrophy”. Please rewrite.

Reply: Thanks for your suggestion. The keyword was replaced by the “muscle fibers”. We hope these revisions can make the keyword more clear and understandable. Thank you so much.

  1. p. 3 line 95-98 Methods 2.4. Not clear what authors mean in line 95-98. Please rewrite.

Reply: Thanks for your suggestion. We have rewritten the method 2.4.

  1. p. 3 line 131. In materials and methods, Hematoxylin-eosin staining was presented as HE staining. However, here it was shown by H&E staining. Please keep the full text consistent with each other.

Reply: Thanks for your question. We have changed the H&E to HE and keep the full text consistent with each other.

  1. p. 6 line 197. N.D should be spelled out.

Reply: Thanks for your question. Abbreviations: N.D. has spelled out in line 204.

Reviewer 2 Report

1.       English expressions are poor, and there are many grammatical issues with the manuscript.

2.       It is better to remove the legend of fig 1B.

3.       Line 169: Chang text “Table 3” to “Table 2”.

4.       Line 206, 208: NEFA or NEAA?

5.       Line 206, 207: Chang text “Eaa” to “EAA”.

Author Response

On behalf of my co-authors, we appreciate you greatly for these positive and constructive comments on our manuscript entitled “Effects of Sex on the Morphological characteristics of Muscle Development and Meat Composition in Wuliangshan Black-bone Chickens” (animals-191706). The comments are all valuable and very helpful for revising and improving our paper, as well as has important guiding significance to our researches. Our manuscript has undergone English language editing by MDPI. We have addressed all the comments point by point, and the revision highlighted in the manuscript by track changes. All response to your comments is listed below.

Reviewer 2

  1. English expressions are poor, and there are many grammatical issues with the manuscript.

Reply: Thanks for your suggestion. Our manuscript has undergone English language editing by MDPI. Spelling and grammatical error were performed in the revised version.

  1. It is better to remove the legend of fig 1B.

Reply: Thanks for your question. But the fig 1B is an important result which is support that muscle types influenced CSA of muscle fibers.

  1. Line 169: Chang text “Table 3” to “Table 2”.

Reply: Thanks for your question. We have changed text “Table 3” to “Table 2”.

  1. Line 206, 208: NEFA or NEAA?

Reply: Thanks for your question. We have changed NEFA to NEAA.

  1. Line 206, 207: Chang text “Eaa” to “EAA”.

Reply: Thanks for your suggestion. We have changed text “Eaa” to “EAA”. 

Reviewer 3 Report

This manuscript titled “Effects of Sex on the Morphological characteristics of Muscle Development and Meat Composition in Wuliangshan Black Bone Chickens” performed the hematoxylin-eosin (HE) to explore the development of breast and thigh muscles in males and females. Then the body weight of cocks and hens have reached 2.0 kg and 1.5 kg weight at 126 days of age, the meat composition of breast and thigh muscles were assessed. This work would provide new insights to better understand the sexual effect on breast and thigh muscle and meat composition in Wuliangshan Black Bone Chickens. However, some issues need be addressed before publications.

Major concerns:

1. The authors compare the difference of breast and thigh muscles between cocks and hens. Although these difference in muscle is associated with gender, the current data cannot support the title and advice changed it into “The sexual difference of Morphological characteristics of Mus-cle Development and Meat Composition in Wuliangshan Black-bone Chickens”.

2.  For figure 1, adding a subtitle “The skeletal muscle development of Wuliangshan Black-bone Chickens” in the result would be interpretation the data better.

3. For the fiber size analysis, the authors showed the average the cross-sectional area (CSA) of

Myofibers and adding the distribution of different size Myofibers would be better.

4. The table in line 189 on page 5 is sorted incorrectly. There are two tables 3.

5. The language of this article need be improved.

Author Response

On behalf of my co-authors, we appreciate you greatly for these positive and constructive comments on our manuscript entitled “Effects of Sex on the Morphological characteristics of Muscle Development and Meat Composition in Wuliangshan Black-bone Chickens” (animals-191706). The comments are all valuable and very helpful for revising and improving our paper, as well as has important guiding significance to our researches. Our manuscript has undergone English language editing by MDPI, meanwhile we have provided the certificate of English editing. We have addressed all the comments point by point, and the revision highlighted in the manuscript by track changes. All response to your comments is listed below.

Reviewer 3

  1. This manuscript titled “Effects of Sex on the Morphological characteristics of Muscle Development and Meat Composition in Wuliangshan Black Bone Chickens” performed the hematoxylin-eosin (HE) to explore the development of breast and thigh muscles in males and females. Then the body weight of cocks and hens have reached 2.0 kg and 1.5 kg weight at 126 days of age, the meat composition of breast and thigh muscles were assessed. This work would provide new insights to better understand the sexual effect on breast and thigh muscle and meat composition in Wuliangshan Black Bone Chickens. However, some issues need be addressed before publications.

Reply: Thanks for your question. Guobiao Zhou et al have indicated that the body weight of cocks and hens have reached 2.0 kg and 1.5 kg weight at 126 days of age (Zhou et al., 2022).

Zhou, G.; Ai, Q.; Yin, W. Determination and Correlation Analysis of Body Weight and Body Measurement of Wuliangshan Black-bone Chickens. Animal husband and food science. 2022, 43(4), 80-83. (Chinese journal)

  1. The authors compare the difference of breast and thigh muscles between cocks and hens. Although these difference in muscle is associated with gender, the current data cannot support the title and advice changed it into “The sexual difference of Morphological characteristics of Muscle Development and Meat Composition in Wuliangshan Black-bone Chickens”.

Reply: Thanks for your suggestion. We have changed the title into “The Sexual Difference of Morphological Characteristics of Muscle Development and Meat Composition in Wuliangshan Black-bone Chickens”.

  1. For figure 1, adding a subtitle “The skeletal muscle development of Wuliangshan Black-bone Chickens” in the result would be interpretation the data better.

Reply: Thanks for your suggestion. We think that the result “Histological Comparisons of Breast and Thigh Muscles in Wuliangshan Black-bone Chickens” include“The skeletal muscle development of Wuliangshan Black-bone Chickens” content, so we haven't adding a subtitle.

  1. For the fiber size analysis, the authors showed the average the cross-sectional area (CSA) of Myofibers and adding the distribution of different size Myofibers would be better.

Reply: Thanks for your suggestion. We just wanted to compare the differences in muscle fiber area, so we haven't statistic the distribution of different size myofibers.

  1. The table in line 189 on page 5 is sorted incorrectly. There are two tables 3.

Reply: Thanks for your question. We have changed text “Table 3” to “Table 2”.

  1. The language of this article need be improved.

Reply: Thanks for your suggestion. Our manuscript has undergone English language editing by MDPI. Spelling and grammatical error were performed in the revised version.
